# Sex-Dependent Effects of Piromelatine Treatment on Sleep-Wake Cycle and Sleep Structure of Prenatally Stressed Rats

**DOI:** 10.3390/ijms231810349

**Published:** 2022-09-08

**Authors:** Jana Tchekalarova, Lidia Kortenska, Pencho Marinov, Natasha Ivanova

**Affiliations:** 1Institute of Neurobiology, Bulgarian Academy of Sciences (BAS), 1113 Sofia, Bulgaria; 2Institute of Information and Communication Technologies, Bulgarian Academy of Sciences (BAS), 1113 Sofia, Bulgaria

**Keywords:** prenatal stress, sex, piromelatine, motor activity, sleep/wake cycle, BDNF

## Abstract

Prenatal stress (PNS) impairs the circadian rhythm of the sleep/wake cycle. The melatonin (MT) analogue Piromelatine (Pir) was designed for the treatment of insomnia. The present study aimed to explore effects of Pir on circadian rhythmicity, motor activity, and sleep structure in male and female rats with a history of prenatal stress (PNS). In addition, we elucidated the role of MT receptors and brain-derived neurotrophic factor (BDNF) to ascertain the underlying mechanism of the drug. Pregnant rats were exposed to different stressors from day seven until birth. Piromelatine (20 mg/kg/day/14 days) was administered to young adult offspring. Home-cage locomotion, electroencephalographic (EEG) and electromyographic (EMG) recordings were conducted for 24 h. Offspring treated with vehicle showed sex-and phase-dependent disturbed circadian rhythm of motor activity and sleep/wake cycle accompanied by elevated rapid eye movement (REM) pattern and *theta* power and diminished non-rapid eye movement (NREM) sleep and *delta* power. While Pir corrected the PNS-induced impaired sleep patterns, the MT receptor antagonist luzindol suppressed its effects in male and female offspring. In addition, Pir increased the BDNF expression in the hippocampus in male and female offspring with PNS. Our findings suggest that the beneficial effect of Pir on PNS-induced impairment of sleep/wake cycle circadian rhythm and sleep structure is exerted via activation of MT receptors and enhanced BDNF expression in the hippocampus in male and female offspring.

## 1. Introduction

Repeated exposure to variable stressors during pregnancy might affect brain development and result in sustainable brain reprogramming in offspring [1,2]. Experimental studies on the model of prenatal stress (PNS), including our recent report, revealed that repeated stress procedures during pregnancy could cause long-term neurobiological alterations in offspring [3,4,5,6]. Prenatally stressed rats showed depressive-like responses with hyperactivity of the hypothalamus–pituitary–adrenal cortex axis (HPA) suggesting that this model resembles depression in humans [3,6,7,8,9,10]. Moreover, sex-dependent behavioral and biochemical differences were observed in the PNS model, with female rats exhibiting heavier concomitant disturbance than male offspring [3,6,7,8,9]. Offspring with a history of PNS have disturbed rhythms of motor activity and sleep/wake cycle [4,5,7]. Clinical studies demonstrated a positive correlation between impaired sleep architecture and depressive behavior [10,11]. Interestingly, exacerbated rapid eye movement (REM) sleep expression was observed both in patients with depression [12,13,14] and in models of depression [15,16], including PNS [4]. Accordingly, the latency for the onset of REM was decreased, while the number of episodes and duration were elevated in patients with depression [12,13,14] and models of depression [4,15,16]. In reverse, non-rapid eye movement (NREM) sleep was decreased, while Wake episodes were prolonged [4,13,15,16,17,18].

The PNS model is promising for testing new compounds for their potential capacity to treat the long-term neurobiological consequences associated with clinical depressive symptoms. In addition, literature data suggest that the expression of the brain-derived neurotrophic factor (BDNF) is suppressed in depression and sleep disturbance [6]. The melatonin (MT) receptor agonist Piromelatine (Pir) (N-(2-[5-methoxy-1H-indol-3-yl]ethyl)-4-oxo-4H-pyran-2-carboxamide) has been recently developed for the treatment of insomnia [19] and with a longer half-life has an advantage over MT [20]. In addition to the agonistic activity on MT_1_/MT_2_ receptors, the compound behaves as an agonist on serotonin 5HT_1A_/_1D_ receptors [19,21]. Recently, we demonstrated that chronic Pir treatment exerts beneficial effects on emotional disturbance via correction of impaired feedback regulation of the HPA axis and glucocorticoid receptors in the hippocampus [3]. However, these effects were sex-dependent and more pronounced in female rats.

Furthermore, the experimental studies focused on the effect of Pir on sleep architecture and the sleep/wake cycle in offspring with PNS is missing. Therefore, in the present study, we aimed to evaluate the effect of sub-chronic treatment with Pir on home-cage motor activity and sleep/wake profile of male and female PNS-offspring rats. In addition, we explored the role of MT receptors and BDNF in the hippocampus to ascertain the mechanism of action of the melatonin analogue in a PNS rat model of depression.

## 2. Results

### 2.1. Effect of Chronic Piromelatine Treatment on 24-h Home Cage Motor Activity in Male and Female PNS-Exposed Rats

Home cage motor activity during a period of 24 h registration was significantly elevated during the active Dark phase in both male and female rats with a history of PNS (male: *p* < 0.001 vs. C-veh; female: *p* = 0.0047 vs. C-veh) (Figure 1A and Figure 2A). However, the treatment with Pir attenuated the PNS-induced elevated locomotion in the two sexes (male rats: *p* < 0.001 vs. PNS-veh; female rats: *p* = 0.0153 vs. PNS-veh) (Figure 1A and Figure 2A). Cosinor analysis and one-way ANOVA demonstrated a circadian rhythmic pattern in male and female vehicle-treated control groups (Figure 1B,C and Figure 2B,C and inset on the right) (Appendix A). Motor activity fluctuation was maintained both in male and female rats with PNS (male: *p* < 0.001; female: *p* = 0.003) (Figure 1B and Figure 2B). The PNS-veh rats showed significantly elevated motor activity vs. C-veh group at *zeitgeber* (ZT) 0,2,3,10,12,14,16,17,18,21,22,23 in male rats and ZT 2,3,13,16,17 in female rats, respectively (Figure 1B and Figure 2B). Preserved circadian rhythm of motor activity was also detected for C-Pir and PNS-Pir male and female groups, respectively (Cosinor analysis: male: *p* < 0.001; female: *p* < 0.001 and *p* < 0.003) (Figure 1C and Figure 2C).

### 2.2. Effect of Chronic Piromelatine Treatment on Sleep/Wake Cycle and Sleep Architecture in Male and Female PNS-Exposed Rats

#### 2.2.1. The PNS-Induced Enhanced Wake Pattern during the Light and the Dark Phase Was Corrected in Male and Female Rats via Melatonin (MT) Receptors

The male and female rats, pretreated with veh/Pir for 14 days were left undisturbed during the 24-h EEG recording. The melatonin receptor antagonist Luzindol was applied before the Pir injection during the last three days before the EEG registration in rats of both sexes (C-Pir-Luz group and PNS-Pir-Luz group, respectively). The criteria for sleep onset expressed by EEG recorded waves of NREM and REM were accepted as in the previous report [16] and according to [22]. We report that neither the PNS model nor drug treatment provoked changes in the latency for NREM and REM onset, respectively (*p* > 0.05) (Table 1 and Table 2). However, the male veh-treated rats with PNS showed an increased total number and duration of Wake events (*p* < 0.01 and *p* = 0.0445, respectively, vs. C-veh rats) (Table 1). The female PNS-veh rats also exhibited prolonged total time of Wake events (*p* = 0.045 vs. C-veh rats) (Table 2).

While in male rats, the Wake events (number and duration) were increased only during the Light phase (number: *p* = 0.0074 vs. C-veh, Figure 3A; duration: *p* = 0.05 vs. C-veh, Figure 4A, Appendix A), Wake pattern (duration) was increased during both the Light (duration: *p* = 0.01 vs. C-veh) and the Dark phase (*p* < 0.001 vs. C-veh) in female PNS-veh rats (Figure 5A, Appendix A).

The treatment with Pir caused a decrease in total Wake events both in male (number: *p* < 0.05 vs. PNS-veh) and in female (duration: *p* < 0.05 vs. PNS-veh) rats with PNS (Table 1 and Table 2). Sub-chronic treatment with Pir decreased the number and duration of Wake events in PNS rats during the Light phase in male rats (*p* < 0.05 vs. PNS-veh) and female rats (duration: *p* < 0.05 vs. PNS-veh) and the Dark phase (duration: male and female: *p* < 0.04 vs. PNS-veh) (Figure 3A, Figure 4A and Figure 5A, Appendix A). Luzindol blocked the effect of Pir on the duration of Wake events during the Light and the Dark phase both in male and female PNS rats (*p* < 0.05 vs. PNS-Pir) (Figure 4A and Figure 5A). Cosinor analysis and one-way ANOVA revealed that the circadian rhythm pattern demonstrated in the C-veh group (male: *p* = 0.002; female: *p* < 0.001) was preserved in male and female controls treated with Pir (male: *p* < 0.001; female: *p* < 0.001) (Figure 4B and Figure 5B and inset on the right; Appendix A). Moreover, the melatonin analogue corrected the PNS-induced impairment of Wake circadian oscillations (male: *p* < 0.001; female: *p* = 0.014) (Figure 4C; Appendix A) (Figure 5C; Appendix A). The PNS elevated the duration of Wake events at ZT0, ZT2, ZT3, and ZT6 in male rats (*p* < 0.05 vs. C-veh) (Figure 4B) and at ZT1, ZT5, ZT7, ZT8, ZT9, ZT10, ZT14 and ZT15 in female rats (*p* < 0.05 vs. C-veh) (Figure 5B).

#### 2.2.2. The Sub-Chronic Treatment with Piromelatine Alleviated the Pns-Induced Reduction of the Nrem Pattern during the Light and the Dark Phase in Male and Female Rats via the Melatonin (Mt) Receptors

Prenatal stress diminished the NREM events both in male rats (total duration: *p* = 0.015 vs. C-veh), and female rats (total number of events: *p* < 0.05 and total duration: *p* = 0.003 vs. C-veh) (Table 1 and Table 2, Figure 6A,B). The observed PNS-induced decrease of NREM pattern was evident during the Light phase in male rats (duration: *p* < 0.001 vs. C-veh) (Figure 7A, Appendix A) and female rats (number of events: *p* = 0.0055 and duration: *p* < 0.001, PNS-veh vs. C-veh) (Figure 6B and Figure 8A, Appendix A; Figure 3A,B).

While Pir did not affect the NREM pattern (number of events) (*p* > 0.05) (Table 1, Figure 6A) this melatonin analogue corrected the PNS-diminished duration of NREM episodes in male controls (total duration: *p* = 0.0473 vs. PNS-veh) (Table 1). This effect was evident during the Light phase (duration: *p* = 0.0242 vs. PNS-veh) (Figure 7A; Appendix A). The drug Pir diminished the latency for onset and the total number of NREM patterns in female control rats (latency: *p* < 0.05, number of events: *p* < 0.01, C-Pir vs. C-veh) (Table 2), and this effect was observed during the Light phase (number of events: *p* = 0.0356 vs. C-veh) (Figure 6B). In contrast, the melatonin analogue attenuated the PNS-diminished duration of NREM pattern during the Light and the Dark phase in female rats (*p* < 0.001, PNS-Pir vs. PNS-veh) (Figure 8A; Appendix A, Figure 3C). Luzindol suppressed the effect of Pir in male and female PNS rats both during the Light and the Dark phase (*p* < 0.001 vs. PNS-Pir) (Figure 7A and Figure 8A). The circadian fluctuations were demonstrated in male and female controls treated with vehicle or Pir (*p* < 0.001), respectively, (Figure 7B,C and inset on the right; Figure 8B,C and inset on the right; Appendix A). Like for the Wake pattern, the PNS-veh group lacked the circadian rhythmicity of NREM sleep (male: *p* = 0.064; female: *p* = 0.063) (Figure 7B and Figure 8B; Appendix A) while Pir corrected it both in male (*p* = 0.004) and in female PNS rats (*p* = 0.016) (Figure 7C and Figure 8C; Appendix A). The duration of the NREM episodes was reduced at ZT7 to ZT8 in male rats (*p* < 0.05 vs. C-veh) (Figure 7B) and at ZT3, ZT4, ZT6, ZT7, ZT8, ZT10, ZT11 in female rats (*p* < 0.05 vs. C-veh) (Figure 8B).

#### 2.2.3. The Sub-Chronic Treatment with Piromelatine Suppressed the Extended Rem Duration Caused by Pns during the Light and the Dark Phase in Male and Female Rats via the Melatonin (Mt) Receptors

Prenatal stress exacerbated the REM sleep in both male (duration of events: *p* < 0.05 vs. C-veh, Appendix A) and female rats (number and duration of events: *p* < 0.05 vs. C-veh) (Table 1 and Table 2, Appendix A). The duration of the REM pattern was increased specifically during the Light phase in male rats (*p* = 0.0286 vs. C-veh) (Figure 9A; Appendix A). However, for female rats with PNS we detected an increased number of events during the Light phase (*p* = 0.0147 vs. C-veh) (Figure 9B). The duration of the REM events was higher both during the Light and Dark phase (*p* = 0.0286 and *p* = 0.002, respectively, vs. C-veh) (Figure 10A, Appendix A).

The treatment with Pir corrected the PNS-induced increased total duration of the REM events (*p* < 0.05 vs. PNS-veh) (Table 1) and this effect was detected both during the Light phase (*p* = 0.0056 vs. PNS-Pir) and the Dark phase (*p* = 0.0007 vs. PNS-Pir) in male rats (Figure 11A, Appendix A). Luzindol blocked the effect of Pir treatment in the PNS male rats during the Light phase (*p* = 0.0034 vs. PNS-Pir) though a tendency for a similar effect was also noticed during the Dark phase without reaching significance (*p* > 0.05 vs. PNS-Pir). The treatment with Pir in PNS rats alleviated the enhanced REM events also in female rats both during the Light phase: (number of events: *p* = 0.021 vs. PNS-veh; duration: *p* = 0.0002 vs. PNS-veh) and the Dark phase: (*p* = 0.0005 vs. PNS-Pir) (Table 2, Figure 9B and Figure 10A, Appendix A). Luzindol blocked the effect of Pir on the PNS-induced increased duration of REM episodes during the Light phase in female rats: (*p* = 0.0022 vs. PNS-Pir) and Dark phase: (*p* = 0.0031 vs. PNS-Pir), respectively, (Figure 10A). The Pir treatment maintained the circadian oscillations as in the control group treated with vehicle in male and female rats (cosinor analysis: male: *p* = 0.006; female: *p* < 0.001) (Figure 10B,C and Figure 11B,C; Appendix A). However, the PNS-impaired diurnal pattern of REM sleep in female rats (*p* = 0.48) was not corrected by Pir (cosinor analysis: female: *p* = 0.81). The duration of the REM episodes was prolonged at ZT3, ZT4, ZT5, ZT6, ZT8 in male rats (Figure 10B) and ZT0, ZT1, ZT3, ZT8, ZT10, ZT12, ZT13, ZT14, ZT15, ZT17, ZT19, ZT21, ZT23 in female rats (Figure 12B).

### 2.3. Effect of Chronic Piromelatine Treatment on BDNF Expression in Male and Female PNS-Exposed Rats

Prenatal stress provoked suppression of BDNF expression in the hippocampus of male rats (*p* = 0.031 vs. C-veh) but not in female rats (*p* > 0.05 vs. C-veh) (Figure 12A,B). However, Pir treatment elevated this growth factor level compared to the PNS-veh group in the two sexes (*p* < 0.05).

## 3. Discussion

The present findings suggest that the novel melatonin analogue piromelatine, developed to correct sleep disturbance in humans, has potential beneficial activity against the PNS-induced impaired sleep/wake cycle and sleep architecture via MT receptor activation and enhanced BDNF expression both in male and in female rats.

The prenatal stress model is characterized by changed emotional responses, including elevated anxiety and depressive-like behavior in male and female rats [23,24]. Recently, we have demonstrated that PNS induced a diminished total activity in the open field test and the elevated plus-maze test both in male and female offspring [3], which is suggested to reflect disturbed emotional responses with enhanced anxiety in a novel environment. In the present study, male offspring with a history of PNS had elevated motor activity in home cages, similar to reported data by Mairesse et al. [4] with PNS female rats. Our results suggest that melatonin receptors might be involved in the beneficial effect of Pir on the PNS-induced changes in home-cage motor activity in male and female rats. The crucial role of the melatonin system in the maintenance of motor activity and its circadian rhythms is discussed earlier in our reports on models of melatonin deficit [16] as well as by other authors with manipulation of melatonin release [25,26].

The harmful impact of PNS on the sleep architecture of rat offspring is well-documented [4,5,27,28]. We found that PNS rats exhibited higher Wake episodes with sex-dependent changes in circadian fluctuations. The enhanced Wake pattern (the number of events and duration) in male PNS-veh rats was limited to the Light phase when the nocturnal naïve rats are suggested to be inactive. Although the number of Wake episodes was not changed in female PNS-veh rats compared to controls, they showed prolonged total Wake duration detected both during the Light and the dark phases, suggesting a more pronounced impairment of wake/sleep cycle in this sex. This result is consistent with clinical data for a higher vulnerability of females to depression [29,30]. Some studies on male PNS rats contradict our results [4,5,27,28]. Dugovic et al. [27] reported fewer Wake episodes during the dark phase. Mairesse et al. [4] showed no difference in Wake episodes, while Sickmann et al. [28] found a sex-dependent decreased number/duration of Wake episodes during the Light and Dark phase in male and female rats. The discrepancy in the reported PNS-induced changes in Wake episodes might be due to divergent strains and the methodology applied to induce PNS.

Furthermore, the findings regarding PNS-induced alterations in sleep architecture among studies are also controversial and not homogenous, revealing diminished NREM patterns and high REM sleep in male rats during the Dark phase [4,5], increased expression of REM sleep during the Light and the Dark phase in male rats [27] and increased NREM in the onset of the Dark phase in the two sexes [28]. Furthermore, we report that PNS induced diminished NREM bout (duration) in male and NREM bout (number and duration) in female rats, respectively, during the inactive Light phase. Contrary to the NREM sleep changes in PNS rats treated with vehicle, REM sleep episodes were elevated during the Light and the Dark phase without sex difference, which agrees with the report of Dugovic et al. [27]. However, Sickmann et al. [28] and Mairesse et al. [4] demonstrated that PNS induced an increase in the REM sleep during the active Dark phase though data in [4] showed a tendency for elevation without reaching a significance of the REM sleep also during the Light phase.

Recently, the novel antidepressant agomelatine, melatonin analog, was reported to correct the PNS-induced disturbed rhythms of motor activity and sleep/week cycle in PNS rats [4]. The chronobiotic properties of agomelatine were demonstrated in different animal models [31,32,33]. Moreover, the beneficial effects shown by agomelatine were attributed to the correction of the impaired melatonin system that might be responsible for disturbed sleep architecture in PNS [4]. To explore the role of melatonin receptors in the beneficial effect of Pir on PNS-induced impairment of circadian rhythm of motor activity and sleep structure, we applied treatment with the potent MT_1_/MT_2_ receptor antagonist luzindol [34]. Piromelatine was designed by Neurim Pharmaceuticals for the treatment of insomnia, including sleep and circadian rhythm disturbance in Alzheimer’s disease. A previous study on mice revealed that Pir could enhance NREM sleep while diminishing the Wake pattern [35]. However, although promising preclinical data were reported in naïve mice, the company Neurim did not develop this melatonin analogue for the treatment of insomnia because of ambiguous clinical results (https://clinicaltrials.gov/ct2/show/results/NCT01489969 accessed on 1 June 2022). In the present study, we report for the first time that Pir possesses chronobiotic activity in a rat model of PNS by restoring the disturbed circadian rhythms of Wake and NREM episodes in female rats. Moreover, we have reported earlier that Pir has an antidepressant potency in PNS [3] considered as a model of depression [5]. Because depression is characterized by a desynchronized circadian rhythms of sleep/wake cycle, including HPA axis hyperactivity [36,37], our previous and present findings support the presumption that Pir might be considered an antidepressant candidate in the PNS model with potency to correct circadian rhythm abnormalities, sleep disturbances and HPA axis hyperactivity through activation of MT receptors. It is also important to notice that the treatment protocol for Pir was adjusted according to the common schedule of treatment with melatonin analogues 1–2 h before the onset of the Dark phase when melatonin levels are raised in human and nocturnal rats [38] and when Pir would activate both MT_1_ and MT_2_ receptors [39]. The activation of these receptors has been shown to be implicated in sleep structure [40]. Moreover, while MT_1_ receptors were suggested to be involved in regulating REM sleep, the MT_2_ receptors were considered responsible for NREM sleep. The selective melatonin receptor antagonist luzindol was reported to exert a higher affinity for the MT_2_ than the MT_1_ receptor [41]. Curiously, we found that it suppressed the effects of Pir on PNS-induced changes in both NREM and REM sleep, suggesting that the two MT_1_ and MT_2_ receptors mediated Pir effects on sleep structure. The BDNF level was unchanged in the PNS female rats (compared to the C-veh groupl) while it was reduced in males. However, the MT receptor agonist Pir enhanced the expression of BDNF in the hippocampus both in male and female rats with a history of PNS. The PNS-induced diminished protein level was restored by Pir in male rats, which agrees with previous reports suggesting that changed BDNF level in the hippocampus is an important mechanism of memory-enhancing activity of Pir treatment [42]. Kushikata et al. [43] reported that NREM sleep could be enhanced by BDNF in rats.

## 4. Materials and Methods

### 4.1. Animals

Young adult offspring Sprague–Dawley rats from the two sexes, weighting 200–250 g (Charles River, Calco, Italy), without (controls) or with a history of PNS, were used. They were accommodated in standard plexiglass cages (four animals per cage), with an ambient temperature of 21 ± 1, humidity 50–60% and maintained in a 12-h artificial light-dark cycle with a light on at 08:00 a.m. The animal had access to water and food ad libitum. The experimental protocol was approved by the Bulgarian Food Safety Agency (#58000183) in full compliance with the European Communities Council Directives of 24 November 1986 (86/609/EEC).

### 4.2. Drugs, Treatment and Experimental Protocol

The melatonin analogue Piromelatine (Pir) (kindly gifted by Neurim Pharmaceuticals Ltd., Tel Aviv, Israel), dissolved in hydroxyethyl cellulose (1%), was administered sub-chronically for 14 days, at a dose of 20 mg·kg^−1^, i.p. The dose and time of injection (two hours before the onset of the dark phase) was determined based on previous studies [3,21,39,44]. Luzindol was injected in the same route at a dose of 30 mg·kg^−1^ as described earlier. The vehicle was applied in the same route in controls (C) and PNS groups, respectively. In combination (Pir + Luz), the antagonist of MT receptors was pretreated 20 min before each injection of Pir the last three days before the 24-h recording of motor activity and EEG registration of wake/sleep condition. Seventy-two rats were allocated in twelve groups (n = 6): male and female rats treated with vehicle (veh)-, Pir, Luz, and combination Luz + Pir, respectively. After home-cage monitoring, each group was EEG recorded for a 24-h period.

### 4.3. Prenatal Stress (PNS) Rat Model and Experimental Groups

The procedure of PNS was performed as described in our previous study [3]. In brief, pregnant female rats were exposed to different stressors as described earlier: two daily stressors a day were applied in a random way during E7–20 with one short-term (forced swimming, crowding, social stressor, wet bedding, restraint stress) and one long-term stressor overnight (light on, fasting, tilted at 45° home cages). The dams bred for control groups were left undisturbed in home cages. Matched pregnant rats, left undisturbed in their home cages, were housed in a separate room. Postnatal day (P) 0 was the day of delivery. For each group, 3 litters were applied as adult offspring in a group of 6 rats to avoid the “litter false-positive effect” on the studied parameters [45].

### 4.4. The 24-h Registration of Home-Cage Motor Activity

The home-cage locomotion of each rat was registered automatically for three days and the average for 24 h was calculated. Rats were adjusted to the new environment in single-housed home-like cages overnight. Data recording started in the morning by simply switching a microcontroller for each cage. Movement events were captured by two pairs of infrared beam detectors placed 6 cm above the bottom, which transversed the cage. Rats crossing as a number of counts were recorded and analyzed by custom software with MySQL database. Average locomotor activity was presented as total counts for 24 h and distribution per hour for 24 h.

### 4.5. Surgery and EEG Recording

The implantation of electrodes was executed as described in our previous study [20] (Tchekalarova et al., 2020). In brief, anesthetized rats with ketamine (80 mg/kg), i.p. and xylazine (20 mg/kg), s.c. were implanted with two stainless steel Teflon-coated electrodes over the left and right frontal cortex (FC). A reference and a ground electrode were inserted above the nasal bone. Two flexible stainless steel electrodes were implanted into the left and right neck muscles for electromyography (EMG) monitoring. All wires were collected to a six-plug female connector (Plastic One MS363/E363/0), attached over the skull through dental acrylic. Cares with antibiotics injection for three days was conducted on all rats with surgery. EEG recording started seven days later. The 24-h monitoring and analysis of sleep/wake cycle was conducted as described earlier [16] by using the Acknowledge software ACK100W (BIOPAC Inc., Goleta, CA, USA) connected to a preamplifier (8213; Pinnacle Technology Inc., Goleta, CA, USA). The EEG and EMG signals were filtered (EEG: high pass 0.5 Hz; low pass 40 Hz; EMG: high pass 80 Hz; low pass 100 Hz). Manual scoring of digitized EEG and EMG traces was carried out over 8-s epochs to quantify the duration and number of sleep/wake episodes during the light and dark phase and the circadian scheduling of sleep/wake states. Wakefulness was scored when the *beta* activity (14.5–40 Hz) was present, and EMG was higher than NREM. NREM was scored when there was ≥50% slow wave high-amplitude *delta* activity (0.5–4 Hz, 250–500 μV) with concomitant depressed EMG activity. REM sleep was determined at 4–8 Hz with fully depressed EMG activity. Mean duration and number of awake, NREM, and REM sleep for the light and dark was calculated. In addition. the total 24-h period of wake, NREM, and REM sleep episodes were calculated. Mean bout duration was defined as the mean duration of each sleep/wake pattern (in minutes) calculated per hour. EEG quantification was performed by using Discrete Fourier transform (DFT), with spectrum summed over 8 s time windows during 24 h of continuous and noise-free segments as reported earlier [16]. In brief, total absolute power, calculated as the sum of squares of all power values contained in a given band and total absolute power of investigated band was analyzed.

### 4.6. Measurement of BDNF Expression in the Hippocampus

At the end of EEG recording, each rat was decapitated with a guillotine following mild anesthesia with CO_2_. The hippocampi were rapidly dissected, inserted in liquid nitrogen and then stored at −20 °C until biochemical analysis by ELISA kit (Antibodies-online GmbH, Aachen, Germany) according to the manufacturer’s instructions.

### 4.7. Statistical Analysis

Data are given as mean ± S.E.M. SigmaStat^®^ software (version 11.0., San Jose, CA, USA)/GraphPad Prism 6 software was applied for statistical analyses. Two-way ANOVA followed by Fisher’s LSD post hoc test was applied separately for male and female rats. A 24-h cosinor model was applied as followed: f (t) = M + A × Cos + Ø (M = mesor; A = amplitude; Ø = acrophase; T = 24 h).

## 5. Conclusions

In the present study, the effect of the novel melatonin analogue Piromelatine was studied on the sleep/wake cycle and sleep structure in male and female young adult rats with a history of prenatal stress. Our results suggest that Pir exerts a beneficial chronobiotic effect on impaired circadian sleep/wake and NREM rhythm, which is sex-dependent and mediated by the MT receptors and BDNF expression in the hippocampus. Furthermore, the Pir treatment effect on PNS-induced enhanced REM expression in the two sexes was also antagonized by luzindol, suggesting the role of MT receptor activation as a mechanism underlying its effect.

## Figures and Tables

**Figure 1 ijms-23-10349-f001:**
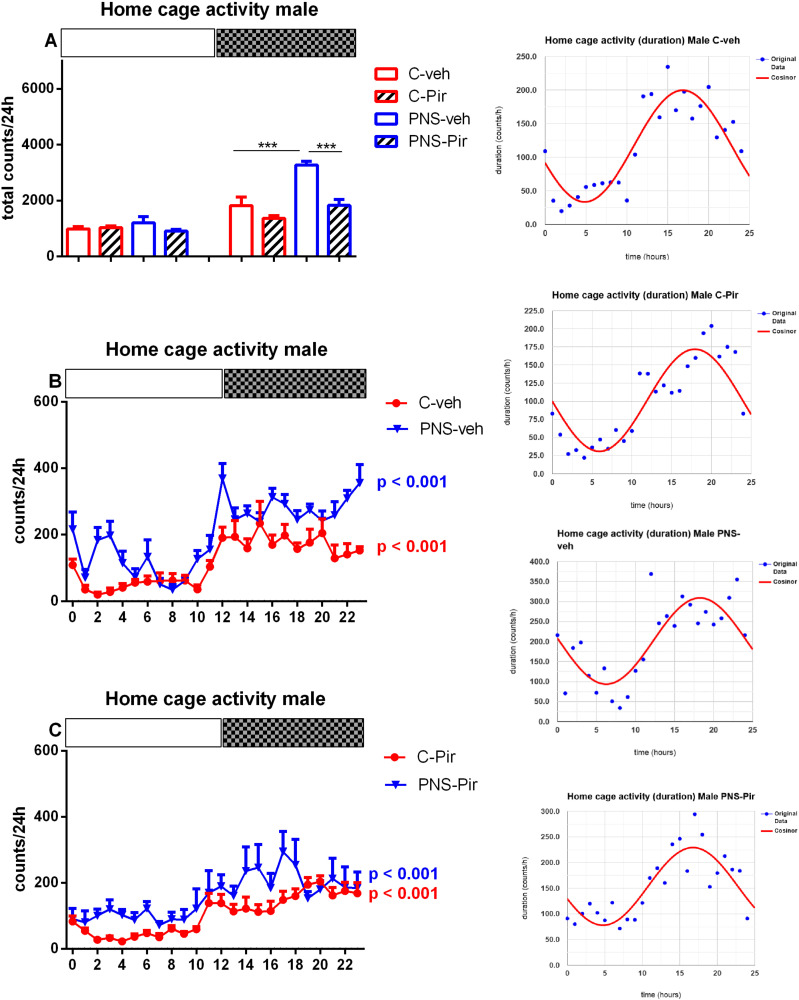
Diurnal variations of home cage activity (total counts) in male rats during the Light and the Dark phase (24-h period registration) for C-veh, C-Pir, PNS-veh and PNS-Pir group (**A**). Light and dark periods are indicated by open and black rectangles above the figure. Data are presented as mean ± SEM, n = 8/group. Circadian distribution of motor activity (counts/h) for C-veh and PNS-veh groups (**B**) and C-Pir and PNS-Pir groups (**C**) over a 24-h recording. Original and cosinor data are shown in the inset on the right. *** *p* < 0.001 vs. C-veh (Dark phase); *** *p* < 0.001 vs. PNS-veh group (Dark phase), respectively.

**Figure 2 ijms-23-10349-f002:**
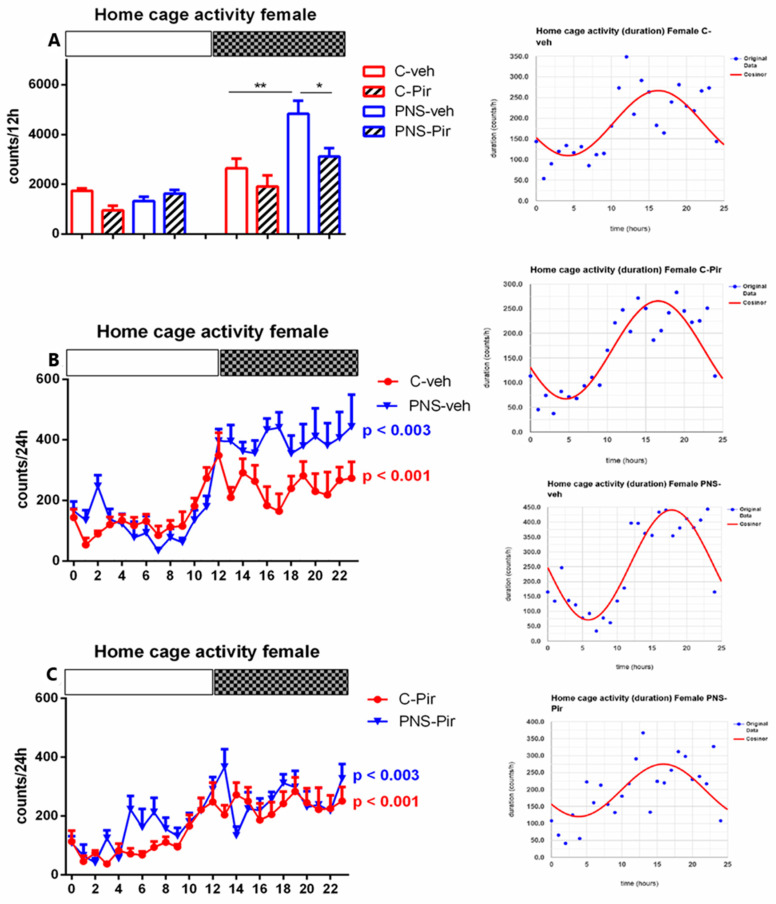
Diurnal variations of home cage activity (total counts) in female rats during the Light and the Dark phase (24-h period registration) for C-veh, C-Pir, PNS-veh and PNS-Pir group (**A**). Light and dark periods are indicated by open and black rectangles above the figure. Data are presented as mean ± SEM, n = 8/group. Circadian distribution of motor activity (counts/h) for C-veh and PNS-veh groups (**B**) and C-Pir and PNS-Pir groups (**C**) over a 24-h recording. Original and cosinor data are shown in the inset on the right. ** *p* = 0.0047 vs. C-veh (Dark phase); * *p* = 0.0153 vs. PNS-veh (Dark phase), respectively.

**Figure 3 ijms-23-10349-f003:**
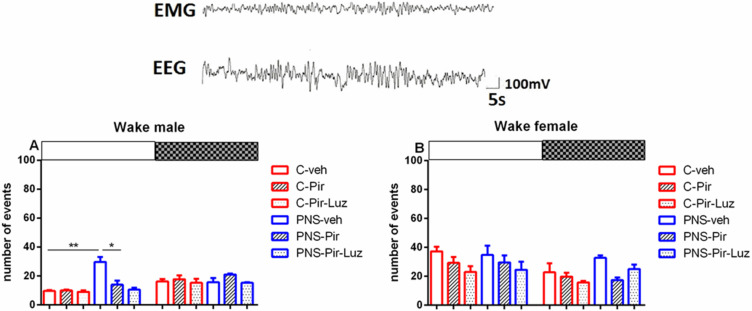
Representative EEG (frontal cortex) and EMG (neck muscle) recording during Wake of the control group. A high-pass filter was set at 0.5 Hz; low pass at 40 Hz. Calibration: 1 s, 100 mV. Diurnal rhythm of Wake number of events during the Light and the Dark phase, indicated by open and black rectangles above the figure, over a 24-h recording in male (**A**) and female rats (**B**) for C-veh, C-Pir, C-Pir-Luz, PNS-veh, PNS-Pir and PNS-Pir-Luz group. ** *p* < 0.01 vs. C-veh (Light phase); * *p* < 0.05 vs. PNS-veh (Light phase), respectively. Details as in Figure 1.

**Figure 4 ijms-23-10349-f004:**
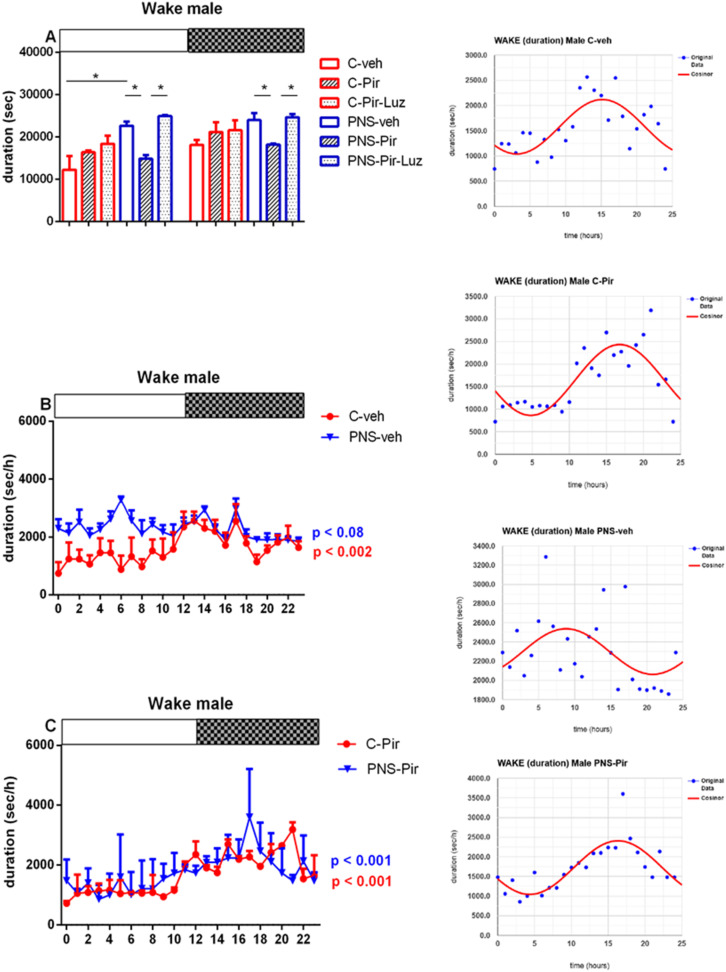
Diurnal rhythm of Wake (duration, sec/h) in male rats during the Light and the Dark phase, indicated by open and black rectangles above the figure, over a 24-h recording for C-veh, C-Pir, C-Pir-Luz, PNS-veh, PNS-Pir and PNS-Pir-Luz group (**A**). Circadian distribution of Wake events (duration, sec/h) for C-veh and PNS-veh groups (**B**) and C-Pir and PNS-Pir groups (**C**) over a 24-h recording. Original and cosinor data are shown in the inset on the right. * *p* < 0.05 vs. C-veh, PNS-veh or PNS-Pir group, respectively. Details as in Figure 1.

**Figure 5 ijms-23-10349-f005:**
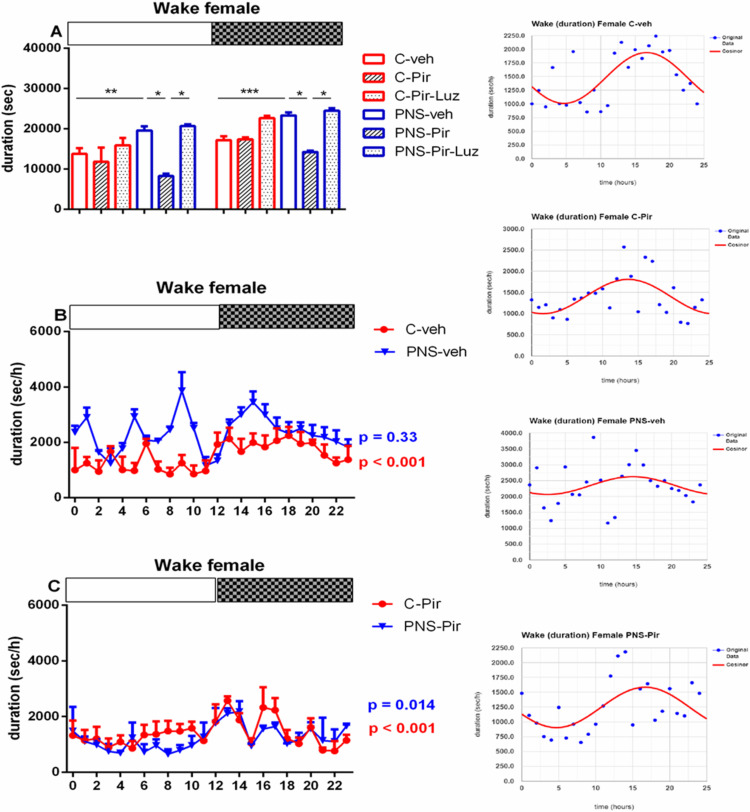
Diurnal wake rhythm (duration, sec/h) in female rats during the Light and the Dark phase, indicated by open and black rectangles above the figure, over a 24-h recording for C-veh, C-Pir, C-Pir-Luz, PNS-veh, PNS-Pir and PNS-Pir-Luz group (**A**). Circadian distribution of Wake events (duration, sec/h) for C-veh and PNS-veh groups (**B**) and C-Pir and PNS-Pir groups (**C**) over a 24-h recording. Original and cosinor data are shown in inset on the right. * *p* < 0.05, ** *p* < 0.01, *** *p* < 0.001 vs. C-veh, PNS-veh or PNs-Pir group, respectively.

**Figure 6 ijms-23-10349-f006:**
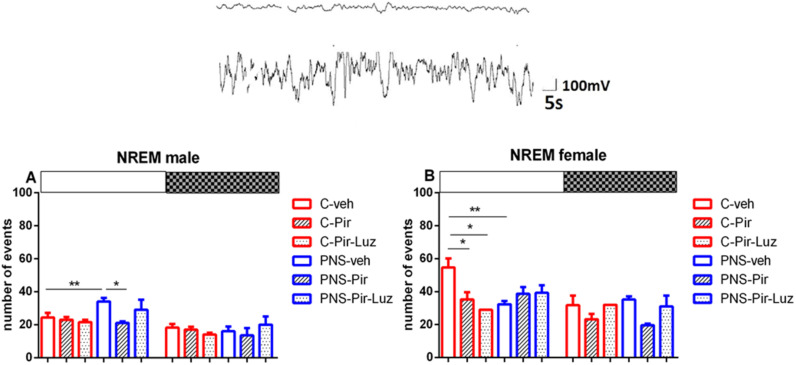
Representative EEG (frontal cortex) and EMG (neck muscle) recording during NREM sleep of the control group. Diurnal rhythm of the number of NREM events in male (**A**) and female rats (**B**) during the Light and the Dark phase, indicated by open and black rectangles above the figure, over a 24-h recording for the C-veh, C-Pir, C-Pir-Luz, PNS-veh, PNS-Pir and PNS-Pir-Luz group. * *p* < 0.05, ** *p* < 0.01.

**Figure 7 ijms-23-10349-f007:**
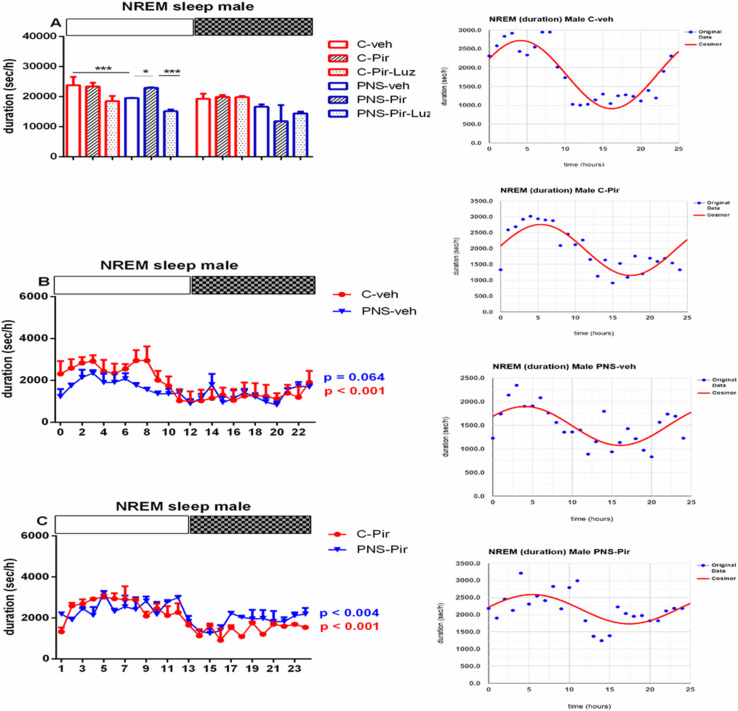
Diurnal rhythm of NREM duration (sec/h) in male rats during the Light and the Dark phase, indicated by open and black rectangles above the figure, over a 24-h recording for C-veh, C-Pir, C-Pir-Luz, PNS-veh, PNS-Pir and PNS-Pir-Luz group (**A**). Circadian distribution of NREM sleep events (duration, sec/h) for C-veh and PNS-veh groups (**B**) and C-Pir and PNS-Pir groups (**C**) over a 24-h recording. Original and cosinor data are shown in inset on the right. * *p* < 0.05, *** *p* < 0.001 vs. C-veh, PNS-veh or PNs-Pir group, respectively.

**Figure 8 ijms-23-10349-f008:**
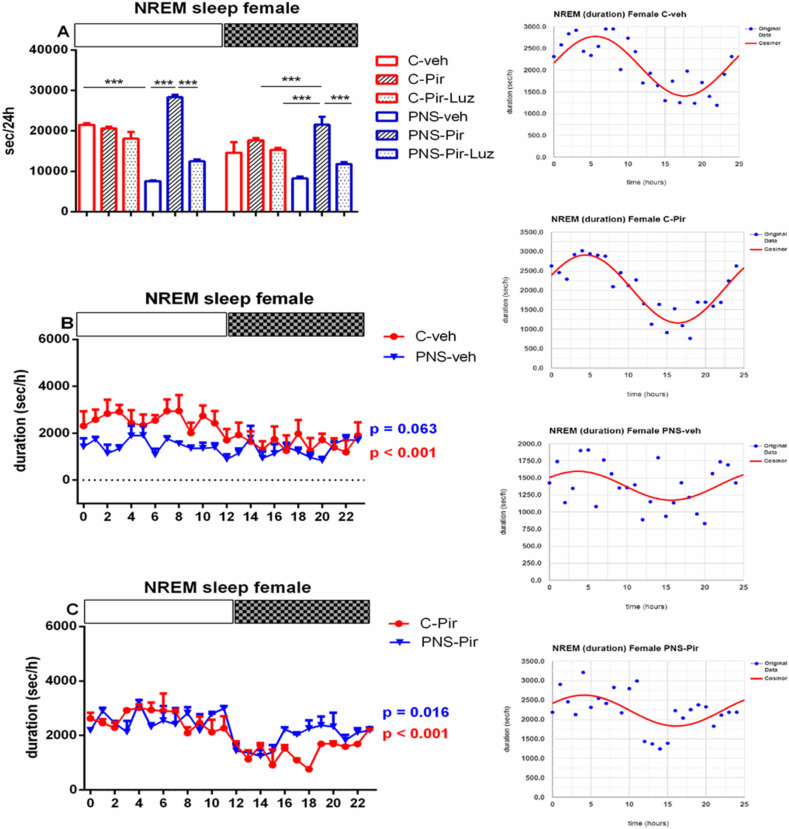
Diurnal rhythm of NREM duration (sec/h) in female rats during the Light and the Dark phase, indicated by open and black rectangles above the figure, over a 24-h recording for C-veh, C-Pir, C-Pir-Luz, PNS-veh, PNS-Pir and PNS-Pir-Luz group (**A**). Circadian distribution of NREM sleep events (duration, sec/h) for C-veh and PNS-veh groups (**B**) and C-Pir and PNS-Pir group (**C**) over a 24-h recording. Original and cosinor data are shown in inset on the right. *** *p* < 0.001 vs. C-veh, PNS-veh or PNS-Pir group, respectively.

**Figure 9 ijms-23-10349-f009:**
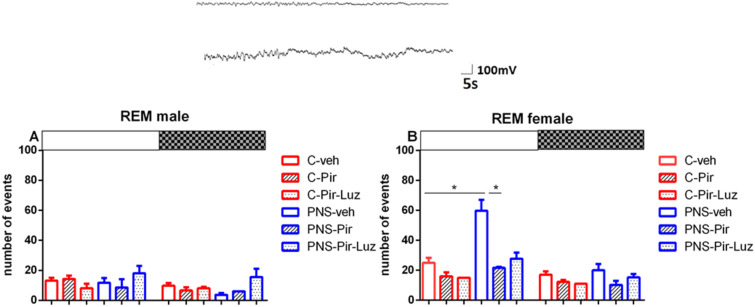
Representative EEG (frontal cortex) and EMG (neck muscle) recording during REM sleep of control. Diurnal rhythm of REM number of events in male (**A**) and female rats (**B**) during the Light and the Dark phase, indicated by open and black rectangles above the figure, over a 24-h recording for C-veh, C-Pir, C-Pir-Luz, PNS-veh, PNS-Pir and PNS-Pir-Luz group. * *p* < 0.05 vs. C-veh or PNS-veh group, respectively.

**Figure 10 ijms-23-10349-f010:**
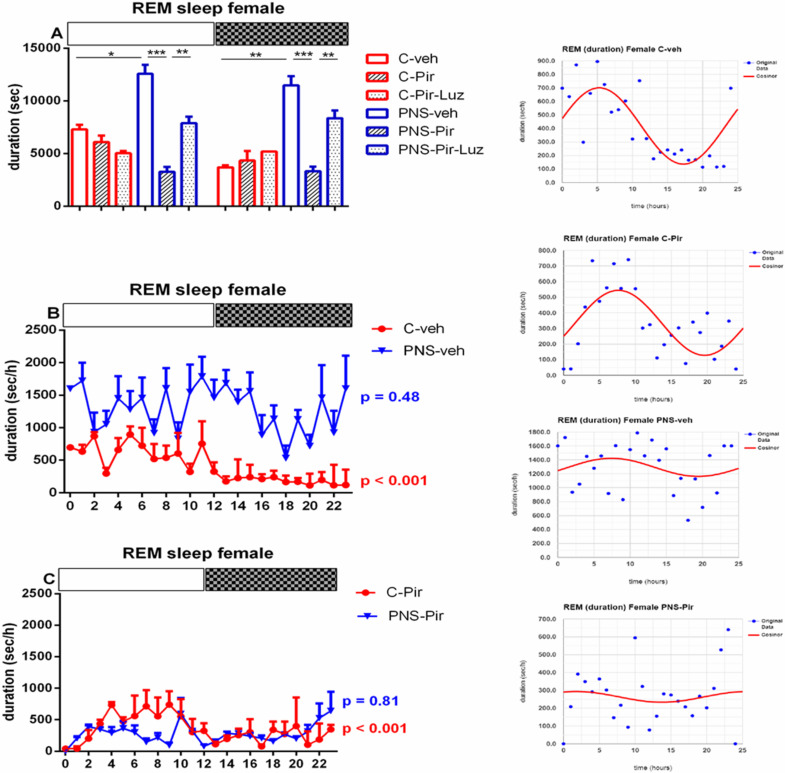
Diurnal rhythm of REM duration (sec/h) in female rats during the Light and the Dark phase, indicated by open and black rectangles above the figure, over a 24-h recording for C-veh, C-Pir, C-Pir-Luz, PNS-veh, PNS-Pir, and PNS-Pir-Luz group (**A**). Circadian distribution of REM sleep events (duration, sec/h) for C-veh and PNS-veh groups (**B**) and C-Pir and PNS-Pir groups (**C**) over a 24-h recording. Original and cosinor data are shown in the inset on the right. * *p* < 0.05, ** *p* < 0.01, *** *p* < 0.001 vs. C-veh, PNS-veh or PNS-Pir group, respectively.

**Figure 11 ijms-23-10349-f011:**
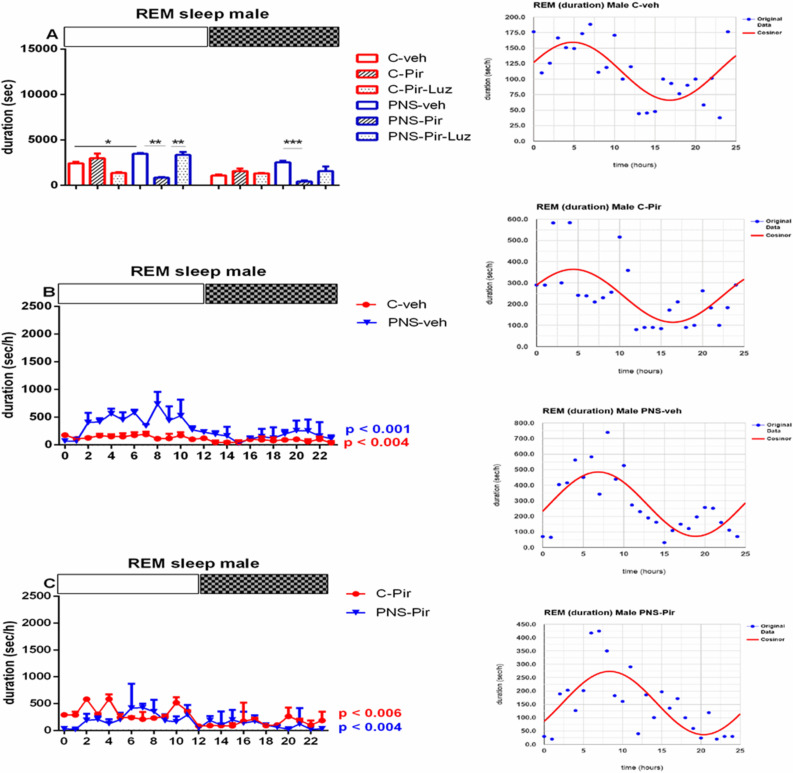
Diurnal rhythm of REM duration (sec/h) in male rats during the Light and the Dark phase, indicated by open and black rectangles above the figure, over a 24-h recording for C-veh, C-Pir, C-Pir-Luz, PNS-veh, PNS-Pir and PNS-Pir-Luz group (**A**). Circadian distribution of REM sleep events (duration, sec/h) for C-veh and PNS-veh groups (**B**) and C-Pir and PNS-Pir group (**C**) over a 24-h recording. Original and cosinor data are shown in the inset on the right. * *p* < 0.05, ** *p* < 0.01, *** *p* < 0.001 vs. C-veh, PNS-veh or PNS-Pir group, respectively.

**Figure 12 ijms-23-10349-f012:**
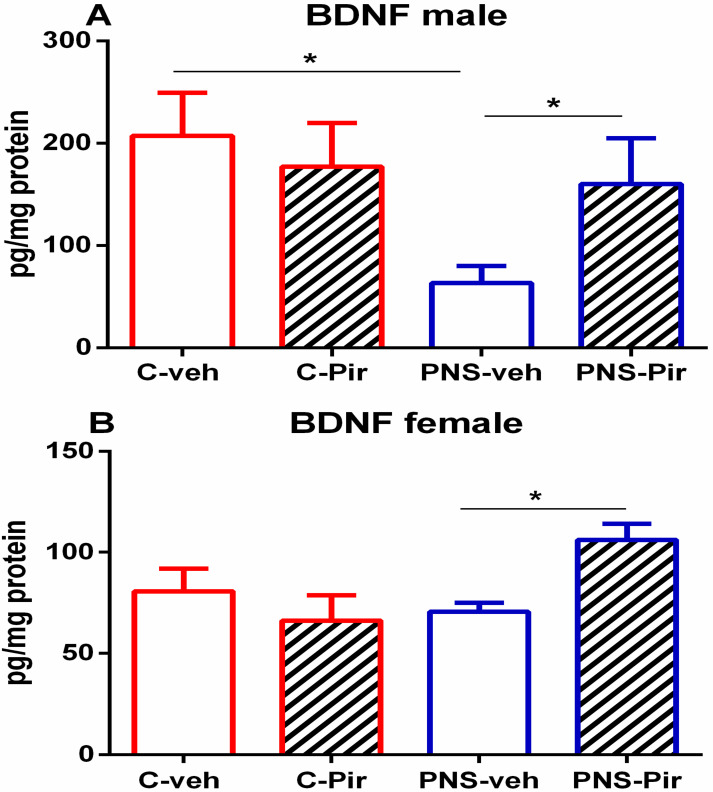
BDNF expression in male (**A**) and female (**B**) C-veh, C-Pir, C-Pir-Luz, PNS-veh, PNS-Pir and PNS-Pir-Luz group in the hippocampus. * *p* < 0.05 vs. C-veh, PNS-veh and PNS-veh group, respectively.

**Table 1 ijms-23-10349-t001:** Male rats: The latency to onset of non-REM (NREM) sleep and rapid eye movement (REM) sleep, number and duration of Wake events, NREM and REM in controls treated with vehicle (C-veh), controls treated with Piromelatine (Pir) (C-Pir), controls treated with Pir + luzindol (Luz) (C-Pir-Luz), rats exposed to prenatal stress (PNS) and treated with vehicle (PNS-veh), rats exposed to PNS and treated with Pir (PNS-Pir) and rats exposed to PNS and treated with Pir + Luz (PNS-Pir-Luz).

Stage	Latency	No of Events	Duration
Wake			
C-veh	NA	25.60 ± 1.80	32,316 ± 520.4
C-Pir		18.00 ± 2.00	38,250 ± 2865
C-Pir-Luz		27.00 ± 3.00	45,410 ± 1352
PNS-veh		45.33 ± 5.36 **	49,975 ± 910.4 *
PNS-Pir		28.11 ± 11.0 ^o^	36,086 ± 281.5
PNS-Pir-Luz		21.50 ± 2.50	50,894 ± 715.5
NREM			
C-veh	76.00 ± 17.00	42.60 ± 3.35	47,203 ± 3586
C-Pir	79.50 ± 8.500	38.50 ± 1.50	43,578 ± 1271
C-Pir-Luz	62.00 ± 13.00	35.50 ± 0.50	38,310 ± 858.2
PNS-veh	63.00 ± 15.37	50.50 ± 4.50	22,581 ± 8029 *
PNS-Pir	72.50 ± 4.500	34.50 ± 3.50	44,507 ± 763.8 ^o^
PNS-Pir-Luz	62.50 ± 19.50	44.50 ± 2.50	30,586 ± 69.86
REM			
C-veh	138.6 ± 23.74	23.00 ± 1.78	5985 ± 1276
C-Pir	161.5 ± 50.50	24.00 ± 7.00	4572 ± 664
C-Pir-Luz	103.5 ± 30.50	16.00 ± 4.00	2681 ± 134.5
PNS-veh	138.7 ± 38.61	15.33 ± 4.09	9500 ± 523.5 *
PNS-Pir	210.00 ± 63.00	14.50 ± 5.50	1292 ± 114.7 ^o^
PNS-Pir-Luz	112.5 ± 12.50	35.50 ± 0.50 ^o^	7410 ± 1478

Values are means (min) per 24 h of recording ± S.E.M. One-way ANOVA followed by Fisher’s LSD post hoc test, * *p* < 0.05, ** *p* < 0.01 vs. C-veh, ^o^
*p* < 0.05 vs. PNS-veh.

**Table 2 ijms-23-10349-t002:** Female rats: The latencies to onset of non-REM (NREM) sleep and rapid eye movement (REM) sleep, number and duration of Wake events, NREM and REM in controls treated with vehicle (C-veh), controls treated with Piromelatine (Pir) (C-Pir), controls treated with Pir + luzindol (Luz) (C-Pir-Luz), rats exposed to prenatal stress (PNS) and treated with vehicle (PNS-veh), rats exposed to PNS and treated with Pir (PNS-Pir) and rats exposed to PNS and treated with Pir + Luz (PNS-Pir-Luz).

Stage	Latency	No of Events	Duration
Wake			
C-veh	NA	60.00 ± 7.29	32,947 ± 1613
C-Pir		49.00 ± 7.01	35,271 ± 1154
C-Pir-Luz		38.50 ± 1.50	44,257 ± 446.5
PNS-veh		67.60 ± 5.40	46,257 ± 1556 *
PNS-Pir		47.00 ± 6.50	26,277 ± 1743 ^o^
PNS-Pir-Luz		51.50 ± 2.50	47,267 ± 1207
NREM			
C-veh	89.00 ± 12.19	86.40 ± 4.94	42,214 ± 1618
C-Pir	40.75 ± 11.40 *	46.00 ± 9.39 **	39,550 ± 1089
C-Pir-Luz	53.50 ± 13.50	61.00 ± 0.00 *	31,906 ± 137.4
PNS-veh	69.60 ± 1.86	66.80 ± 3.99 *	16,145 ± 640.1 **
PNS-Pir	77.33 ± 14.44	58.33 ± 3.75	54,215 ± 1147 ^o^
PNS-Pir-Luz	76.00 ± 10.41	70.25 ± 10.14	23,534 ± 529.9
REM			
C-veh	144.4 ± 15.52	42.00 ± 4.39	10,967 ± 1524
C-Pir	179.5 ± 23.18	28.25 ± 3.09 *	11,104 ± 1059
C-Pir-Luz	113.00 ± 40.00	26.00 ± 0.00	10,238 ± 208.5
PNS-veh	85.20 ± 8.817 *	79.80 ± 11.38 *	23,798 ± 1673 *
PNS-Pir	89.33 ± 12.41	32.00 ± 3.21 ^o^	6670 ± 889.8 ^o^
PNS-Pir-Luz	106.00 ± 13.00	45.50 ± 5.26 ^o^	15,600 ± 997.1

Values are means (min) per 24 h of recording ± S.E.M. One-way ANOVA followed by Fisher’s LSD post hoc test, * *p* < 0.05, ** *p* < 0.01 vs. C-veh, ^o^
*p* < 0.05 vs. PNS-veh.

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
