# Peer review of "Sex-Dependent Effects of Piromelatine Treatment on Sleep-Wake Cycle and Sleep Structure of Prenatally Stressed Rats"

_ijms, 2022, doi:10.3390/ijms231810349_

Round 1

Reviewer 1 Report

No extra recommendation.

Author Response

Thank you for carefully evaluating our manuscript entitled “Sex-dependent effects of piromelatine treatment on sleep-wake cycle and sleep structure of prenatally stressed rats”. We have revised the manuscript taking into account the suggested modifications. Track changes highlight all changes in the MS.

Reviewer 2 Report

This is an interesting study on the potential use of the drug Piromelatine to overcome sleep-wake cycle  disturbances that may be associated with depression. The rat model used in these study is appropriate and its limitations are stated in the paper.

The authors should carry out several  corrections to improve the manuscript

Page1Line32 Error in citation style

Page2 line 52 Grammar improvement needed ('the advantage')

Page2 line 70 What is the difference  between PNS male and female with a 'history' of PNS

Page 2 line 78 Define ZT

Page2 line 84 there are two "A" and there is no "C" in the figure . Also, why is A calculated over 12 hours and B and C over 24 hours

Page 2 line 88 Confusing , there are 3 figures and 4 insets Presumably there two insets for B and two for C (this is repeated for several other figures and should be made more clear),Also note that in this figure  the last inset  is labelled REM

Page4 Line 95 The figures are labelled B,A,B rather than A,B,C

Page 5 Line  107 Grammar: rats that pretreated

Page 5 line 108 It would be useful to remind that Luzindol is an melatonin receptor antagonist (it was mentioned in the abstract but not in the introduction)

Page 5 line 112 It is not clear what is meant by latent period: if it is the latency in the table then it cannot refer to the Wake period, otherwise the table does show significant differences

Page 5 line 118 PNS-Pir-Luz duration is 50894 +/- 715 and is reported as not significantly different while PNS-veh 49975+/-910 is significantly different from C-veh (32316 +/- 5204) It does not seem to be right.

Page 7 line163. In figure 4A there is a significant bar covering the the four right columns From the text it appears that there is significant difference of each of the three column to the far right column (C- veh) The conventional way of expressing that is to have three individual significant bars or to have a star over each of the column.

Page 7 line 167 One of the inset is labelled as REM while the others are WAKE duration

Page10 line 187 In figure 6 the Y axis are labelled differently while they represent the same measurement

Page12 line 225 The text of the section title is not clear

Page 16  line281 As explained above the significance bar in the male for the three left column should be changes to individual stars.

Page17 line 312  Remove 'to' and change to 'our results'

Page17  line 314 Difference (spelling)

Page 18 line 363  The authors should mention that that BDNF was unchanged in PNS female (compared to control) while it was reduced in males.

page 18 line 392 Some details of the procedure should be given because the reference for the protocol is not open access.

page 19 line 398 More details should be given to be able to understand the procedure and the figures in this paper. i.e explain 'counts' and 'duration'

Author Response

Dear Editor and Reviewers,

Thank you for carefully evaluating our manuscript entitled “Sex-dependent effects of piromelatine treatment on sleep-wake cycle and sleep structure of prenatally stressed rats”. We have revised the manuscript taking into account the suggested modifications. Track changes highlight all changes in the MS.

Reviewer 2:

Point #1: Page1 Line32 Error in citation style.

Response: Corrected.

Point # 2: Page2 line 52 Grammar improvement needed ('the advantage').

Response: Thank you for this note. The text was improved.

Point # 3: Page2 line 70 What is the difference between PNS male and female with a 'history' of PNS.

Response: In the new version of the manuscript, we deleted “PNS” before male and female rats. Now the text sounds: “…male and female rats with a history of PNS”.  The effect of sex on PNS-induced changes was explored separately for male and female rats. Therefore, the figures give each parameter's results independently for male and female groups.

Point # 4: Page 2 line 78 Define ZT.

Response: We’re thankful for this note and inserted a definition of ZT when it was mentioned for the first time (page 2).

Point # 5: Page2 line 84 there are two "A" and there is no "C" in the figure . Also, why is A calculated over 12 hours and B and C over 24 hours.

Response: The technical mistake regarding the labeling of y-axis labeled “counts/12 h” was corrected to “total counts/24 h”. The letters of figure were corrected to A,B,C.

Point # 6: Page 2 line 88 Confusing , there are 3 figures and 4 insets Presumably there two insets for B and two for C (this is repeated for several other figures and should be made more clear),Also note that in this figure  the last inset  is labelled REM.

Response: The 3 figures show statistical data, while the 4 insets show Cosinor data. Each set relates to one of the 4 groups: C-veh, C-Pir, PNS-veh and PNS-Pir. “REM” was changed to “Home cage activity”.

Point # 7: Page4 Line 95 The figures are labelled B,A,B rather than A,B,C.

Response: The letters of figure were corrected to A,B,C.

Point # 8: Page 5 Line  107 Grammar: rats that pretreated

Response: Thank you for the relevant note. This technical error was corrected.

Point # 9: Page 5 line 108 It would be useful to remind that Luzindol is an melatonin receptor antagonist (it was mentioned in the abstract but not in the introduction)

Response: Following the Reviewer's advice, we mentioned in the Introduction the mechanism of Luzindol as a selective antagonist of MT receptors.

Point # 10: Page 5 line 112 It is not clear what is meant by latent period: if it is the latency in the table then it cannot refer to the Wake period, otherwise the table does show significant differences.

Response: Latent period was changed to latency and Wake was removed (page 8, line 4).

Point # 11: Page 5 line 118 PNS-Pir-Luz duration is 50894 +/- 715 and is reported as not significantly different while PNS-veh 49975+/-910 is significantly different from C-veh (32316 +/- 5204) It does not seem to be right.

Response: First, we noticed a technical error for the SEM of group C-veh (Table 1, Duration) and corrected 32316 ± 5204 to 32316 ± 520.4. The group PNS-Pir-Luz was compared to PNS-veh only and statistical analysis did not indicate a significant difference between PNS-veh (49975 ± 910) and PNS-Pir-Luz (50894 ± 715).

Point # 12: Page 7 line163. In figure 4A there is a significant bar covering the the four right columns From the text it appears that there is significant difference of each of the three column to the far right column (C- veh) The conventional way of expressing that is to have three individual significant bars or to have a star over each of the column.

Response: In figure 4A the text and the figure indicated that there is a significant difference between C-veh and PNS-veh group but not between C-veh and the other two control groups injected by either Pir or Pir-Luz, respectively. We use a line extending between the two groups with a significant difference and the asterisk inserted above the line. If there were a significant difference between C-veh and C-Pir or C-veh and C-Pir-Lus, respectively, we would insert separate line/s with an asterisk above the two groups. We prefer to use this approach for indicating a significant difference between two groups when there are too many groups and to avoid using many different symbols for comparison.

Point # 13: Page 7 line 167 One of the inset is labelled as REM while the others are WAKE duration.

Response: REM changed to WAKE.

Point # 14: Page10 line 187 In figure 6 the Y axis are labelled differently while they represent the same measurement.

Response: Corrected. The same was done for Figure 9AB. We are thankful for this remark.

Point # 15: Page12 line 225 The text of the section title is not clear.

Response: We agree with the Reviewer's remark and modified the text to the section title accordingly.

Point # 16: Page 16  line281 As explained above the significance bar in the male for the three left column should be changes to individual stars.

Response: The response is the same as for Point # 12.

Point # 17: Page17 line 312  Remove 'to' and change to 'our results'.

Response: The text was corrected.

Point # 18: Page17  line 314 Difference (spelling).

Response: Corrected.

Point # 19: Page 18 line 363  The authors should mention that that BDNF was unchanged in PNS female (compared to control) while it was reduced in males.

Response: We’re thankful to the Reviewer for this remark. The hole paragraph was edited accordingly.

Point # 20: page 18 line 392 Some details of the procedure should be given because the reference for the protocol is not open access.

Response: Following the Reviewer's advice we inserted details for the PNS procedure.

Point # 21: page 19 line 398 More details should be given to be able to understand the procedure and the figures in this paper. i.e explain 'counts' and 'duration'.

Response: We’re thankful to the Reviewer for this remark and edited the text.

Reviewer 3 Report

In the present article, the authors have investigated the potential of piromelatine to correct PNS-induced impaired sleep disturbances and have argued that this compound has the ability to rescue these impairments via the MT receptors and enhanced BDNF production. The article is not well-written and needs a lot of editing before it is suitable for the review process (please see some suggested edits below). There also some major concerns about how the data was plotted and data not agreeing between different panels. At this point, I would not recommend this article for publication.

Major:

1)    How is the average counts/12 hr in Fig 1A about 1000 while in the line trace the activity levels seem so low (~100)? Similarly, for other groups too. Why is the y-axis labeled counts/24 hr in some cases while other figures show /12 hr?

2)    Line 136 and 137 need correction. The pattern is the not the same as males during the light phase and only during the dark phase.

3)    Fig 5B doesn’t agree with 5A.PNS-veh is higher in wake during dark phase in A but in B they are overlapping. Why?

Minor:

1)    X-axis don’t have units (hrs).

2)    Figures have mis-labeled panels. E.g. Fig.1 has no C in it. Fig. 2 starts with B and not A.

3)    English needs a lot of correction.

4)    Line 183 – change 6AB to 6A, 6B.

5)    Panels in the same figure have different labels – 6A and 6B, 9A and 9B for example. 

Author Response

Major:                                                                                                                                                                     

Point # 1: How is the average counts/12 hr in Fig 1A about 1000 while in the line trace the activity levels seem so low (~100)? Similarly, for other groups too. Why is the y-axis labeled counts/24 hr in some cases while other figures show /12 hr?

Response: Fig 1A shows total counts for 24 h (for the light and the dark phase), which is e.g. about 1700 for the C-veh group during the light phase, while the trace line in fig 1B shows the activity for every hour, which is about 100 counts/h.  The technical mistake regarding the labeling of y-axis labeled “counts/12 h” was corrected to “total counts/24 h”.

Point # 2: Line 136 and 137 need correction. The pattern is the not the same as males during the light phase and only during the dark phase.

Response: The text was edited according to the discussed results.

Point # 3: Fig 5B doesn’t agree with 5A.PNS-veh is higher in wake during dark phase in A but in B they are overlapping. Why?

Response: In Figure 5A, different groups' effects and comparative analysis were conducted within Light and Dark phases, respectively. Cosinor analyzed the circadian rhythms data in Figures 5B and C. Following the note in Point #3, we checked the data of the female group PNS-veh and compared the Light vs the Dark phase. There was a tendency for higher Wake duration during the Dark phase vs the Light phase but a lack of a significant difference.

Minor:

1)    X-axis don’t have units (hrs).

Response: For the total counts/events or duration, we have indicated Light and Dark periods by open and black rectangles above the figure, mentioned and added in the text of each figure.

2)    Figures have mis-labeled panels. E.g. Fig.1 has no C in it. Fig. 2 starts with B and not A.

Response: Thank you for this remark. These technical mistakes were corrected in both figures (1 and 2).

3)    English needs a lot of correction.

Response: An editing program was used to improve the language.

4)    Line 183 – change 6AB to 6A, 6B.

Response: Changed.

5)    Panels in the same figure have different labels – 6A and 6B, 9A and 9B for example.

Response: Figures 6 and 9 are panels that consist of two separate figures A for male and B for female rats.

Round 2

Reviewer 3 Report

I am satisfied with the edits that the authors have done. Thank you!